# Propagation of SARS-CoV-2 in a Closed Cell Culture Device: Potential GMP Compatible Production Platform for Live-Attenuated Vaccine Candidates under BSL-3 Conditions?

**DOI:** 10.3390/v15020397

**Published:** 2023-01-30

**Authors:** Stephan Klessing, Antonia Sophia Peter, Kirsten Fraedrich, Jannik T. Wagner, Mirko Kummer, Janina Deutschmann, Philipp Steininger, Hans-Dieter Steibl, Klaus Überla

**Affiliations:** 1Institute of Clinical and Molecular Virology, Universitätsklinikum Erlangen, Friedrich-Alexander-Universität Erlangen-Nürnberg, 91054 Erlangen, Germany; 2Department of Dermatology, Universitätsklinikum Erlangen, Friedrich-Alexander-Universität Erlangen-Nürnberg, 91054 Erlangen, Germany; 3Deutsches Zentrum Immuntherapie (DZI), 91054 Erlangen, Germany; 4Miltenyi Biotec B.V. & Co. KG, 51429 Bergisch Gladbach, Germany

**Keywords:** SARS-CoV-2, CliniMACS Prodigy^®^, GMP, virus propagation, live-attenuated vaccines, human challenge study

## Abstract

Live-attenuated SARS-CoV-2 vaccines present themselves as a promising approach for the induction of broad mucosal immunity. However, for initial safety assessment in clinical trials, virus production requires conditions meeting Good Manufacturing Practice (GMP) standards while maintaining biosafety level 3 (BSL-3) requirements. Since facilities providing the necessary complex ventilation systems to meet both requirements are rare, we here describe a possibility to reproducibly propagate SARS-CoV-2 in the automated, closed cell culture device CliniMACS Prodigy^®^ in a common BSL-3 laboratory. In this proof-of-concept study, we observed an approximately 300-fold amplification of SARS-CoV-2 under serum-free conditions with high lot-to-lot consistency in the infectious titers obtained. With the possibility to increase production capacity to up to 3000 doses per run, this study outlines a potential fast-track approach for the production of live-attenuated vaccine candidates based on highly pathogenic viruses under GMP-like conditions that may contribute to pandemic preparedness.

## 1. Introduction

After the emergence of the severe acute respiratory syndrome coronavirus type 2 (SARS-CoV-2) in late 2019, the virus spread rapidly causing a worldwide pandemic with over 621 million confirmed cases and over 6.5 million deaths as of now [1]. To combat this global crisis, several different kinds of SARS-CoV-2 vaccines were developed and licensed at an unprecedented speed [2,3]. Initially, these vaccines proved to be highly effective in preventing SARS-CoV-2 infection and severe illness and death [4,5,6,7]. However, in a phase of waning immunity, breakthrough infections have been reported [8,9,10,11,12,13,14]. These numbers further increased with the emergence of new virus variants, especially the latest variant of concern (VOC), Omicron. Nevertheless, it was found that while more vaccinated people were susceptible to infection by this VOC, the risk of severe illness and death was still strongly decreased [15,16,17,18]. 

A promising approach to improving and broadening mucosal immunity against SARS-CoV-2 is the generation of live-attenuated vaccines based on mutants of SARS-CoV-2 [19,20,21,22,23]. The development of live-attenuated SARS-CoV-2 vaccines requires the identification of mutants that are attenuated sufficiently to be safe, but at the same time replicate to levels that are high enough to induce protective immune responses. While animal studies may provide the first indications of the safety and efficacy of particular live-attenuated vaccine candidates, ultimately, first-in-man clinical studies need to be performed. As SARS-CoV-2 is classified as a biosafety level 3 (BSL-3) organism, live-attenuated vaccine candidates are initially also classified as BSL-3 agents. Downgrading the biosafety level is possible once sufficient safety data are available. However, since collecting the safety data may take months to years, there is a need for a reproducible production system of such live-attenuated vaccines in a biosafety level 3 laboratory under conditions meeting the requirements of Good Manufacturing Practices (GMP). Adherence to both BSL-3 and GMP conditions results in different ventilation requirements that are difficult to reconcile [24]: BSL-3 laboratories must use controlled air flow to ensure that air flows from non-laboratory areas (such as the hallway) into laboratory areas. For GMP laboratories, usually an air pressure gradient away from the medicinal product is used to ensure the purity of the product. In addition, biosafety measures are needed to protect the personnel working in BSL-3 laboratories. Similar hurdles are met by attempts to generate virus stocks for human challenge studies as a possibility to investigate the efficacy of prophylactic or therapeutic strategies against BSL-3 pathogens [25]. Although these requirements can be implemented by complex room-within-a-room constructions, such buildings are not available in Germany to the best of the authors’ knowledge. An alternative production process is therefore needed to be able to consider the development of live-attenuated vaccines against BSL-3 pathogens.

In the present study, we, therefore, evaluated the concept of producing stocks of SARS-CoV-2 in a BSL-3 laboratory using an automated closed cell culture device. When combined with the filling of the bulk challenge virus preparations in a compounding aseptic containment isolator also located in the BSL-3 laboratory, it should be possible to produce stocks of live-attenuated SARS-CoV-2 vaccine candidates or challenge viruses.

## 2. Materials and Methods

### 2.1. Propagation of Serum-Free Vero-E6 Cells

Vero-E6 cells of WHO origin (ECACC no 88020401) were obtained from Nuvonis (Vienna, Austria) and cultured according to the provided instructions. Briefly, 7.5 × 10^6^ cells were thawed, resuspended in 12 mL equilibrated OptiPRO™ (ThermoFisher Scientific, Waltham, MA, USA), supplemented with 4 mM GlutaMax (ThermoFisher Scientific, Waltham, MA, USA) and added to a T175 flask containing 88 mL of supplemented OptiPRO™. After 24 h incubation at 37 °C at 5% CO_2_ and 95% relative humidity, media was replaced with 50 mL equilibrated supplemented OptiPRO™. The passaging cells were washed once with 13 mL of PBS before adding 5 mL of TrypLE (ThermoFisher Scientific, Waltham, MA, USA) to detach cells. Detached cells were resuspended in 15 mL OptiPRO™ and seeded at 40,000 cells/cm^2^ for a 3-day passage. After two additional passages, cells were frozen at 4.5 × 10^6^ cells/vial in 1 mL of OptiPRO™ containing 7.5% DMSO in a Mr.Frosty freezing container (ThermoFisher Scientific, Waltham, MA, USA) at −80 °C and stored until further use.

### 2.2. Generation of Serum-Free SARS-CoV-2 Virus Stocks

The SARS-CoV-2 strain used for inoculation was isolated from a COVID-19 patient in Erlangen (hCoV-19/Germany/ER1/2020; CoV-ER1, GISAID: EPI_ISL_610249), as described previously with some minor alterations (Supplementary Information of [26]). The virus was passaged once on Vero-E6 cells in serum-free OptiPRO™ and the supernatant of the passaged virus was filtered through a 0.45 µm cellulose acetate membrane filter prior to storage at −80 °C.

### 2.3. Propagation of Attenuated Influenza Virus and SARS-CoV-2 in a Closed Cell Culture Device

Fluenz^®^ Tetra vaccine for nasal application (AstraZeneca GmbH, Hamburg, Germany) was purchased at a listed concentration of 10^7^ ± 0.5 FFU per strain and dose (200 µL). The propagation of the attenuated influenza virus vaccine and the SARS-CoV-2 virus, generated as described above under potentially GMP-compatible conditions, was performed in the CliniMACS^®^ Prodigy (Miltenyi Biotec, Bergisch Gladbach, Germany) in the Adherent Cell Culture module (ACC). For this, the tubing set TS730 (Figure A1) was installed according to the manufacturer’s instructions, followed by an integrity check. Sterile CliniMACS^®^ PBS/EDTA (Miltenyi Biotec, Bergisch Gladbach, Germany) and OptiPRO™ SFM (ThermoFisher Scientific, Waltham, MA, USA) bags were attached to the tubing by sterile tube welding with a TSCD-II welder (Terumo, Shibuya, Japan). Tubing was rinsed once with PBS/EDTA, followed by coating of cell culture chamber with 30 mL of 5 µg/mL Biolaminin 521MX (BioLamina, Sundbyberg, Sweden) for 3 h at 39 °C (corresponding to 37 °C in the cell culture chamber). After aspiration of the coating solution, 15 mL of preheated supplemented OptiPRO™ was injected into the cell culture unit (CCU) through the coating module. Without aspiration, 4.5 × 10^6^ freshly thawed Vero E6 cells in 25 mL preheated supplemented OptiPRO™ were added through the inoculation module and rinsed with 25 mL preheated supplemented OptiPRO™. Cells were cultured for 24 h at 39 °C (corresponding to 37 °C in the CCU) at 5% CO_2_ with a picture taken every 12 h, before media was changed to 30 mL fresh supplemented OptiPRO™ the next day, followed by another 24 h of cultivation. For the infection, 1.667 × 10^6^ FFU (33.3 µL) of Fluenz^®^ Tetra vaccine, or 500 µL of the previously generated SARS-CoV-2 stock (3 × 10^5^ TCID_50_/mL), were diluted in 25 mL supplemented OptiPRO, of which 15 mL were added to the CCU via the coating module. After 4 h at 35 °C for Fluenz^®^ Tetra, or 39 °C (corresponding to 33 °C, or 37 °C in the CCU, respectively) and 5% CO_2_ media was changed to 30 mL fresh supplemented OptiPRO™ and a sample was collected from the aspirated supernatant with a sampling adapter (Miltenyi Biotec, Bergisch Gladbach, Germany) and stored at −80 °C until further use. Subsequently, cells were cultured for 24 h at 39 °C (corresponding to 37 °C in the CCU) at 5 % CO_2_ with a picture taken every 12 h, before media was changed to 30 mL fresh supplemented OptiPRO™ the next day, with a sample taken and stored. This was repeated for a total of 5 days, before the tubing set was dismounted according to the manufacturer’s instructions.

### 2.4. qPCR Analysis

RNA of all supernatants was extracted with the QIAamp^®^ Viral RNA extraction kit (Qiagen, Hilden, Germany) according to the manufacturer’s instructions. Subsequent detection of viral RNA encoding for the SARS-CoV-2 RdRP gene was performed according to the previously described real-time reverse transcription PCR by Corman et al. [27]. For this, 12.5 µL Ambion 2× RT-PCR Buffer and 7.5 µM RdRp_fwd primer (5′-GTGARATGGTCATGTGTGGCGG-3′), 10 µM RdRp_rev primer (5′-CARATGTTAAASACACTATTAGCATA-3′) and 2.5 µM RdRp probe (FAM-5′-CAGGTGGAACCTCATCAGGAGATGC-3′-BBQ) were mixed and adjusted to a total volume of 19 µL. After the addition of 1 µL Ambion 25 × RT-PCR Enzyme mix and 5 µL purified RNA for each sample, thermal cycling was performed at 50 °C for 15 min, followed by 95 °C for 10 min and 40 cycles of 95 °C for 8 s and 60 °C for 34 s.

Influenza A and B viral RNA was detected by using the AgPath-ID One-Step RT-PCR Kit (ThermoFisher Scientific, Waltham, MA, USA). IVA primer set was composed of the following oligonucleotides: 5′-AGATGAGTCTTCTAACCGAGGTCG-3′ (forward primer); 5′-TGCAAAAACATCTTCAAGTCTCTG-3′; and 5′-TGCAAAGACATCTTCCAGTCTCTG-3′ (reverse primers) in combination with the probe FAM-5′-TCAGGCCCCCTCAAAGCCGA-3′-BHQ-1. For IVB detection 5′-CCCTGCTTGCTCGWAGYATGG-3′ was used as forward primer, 5′-TGCTTATGGAAGMCACTTTG-3′ as reverse primer and VIC-5′-CGTTGTTAGGCCCTCTGTGGCGA-3′-BHQ-1 as probe. Thermal cycling was performed at 50 °C for 15 min, followed by 95 °C for 10 min and 40 cycles of 95 °C for 8 s and 60 °C for 34 s. A serial dilution of purified RNA of Fluenz^®^ Tetra stock (5 × 10^7^ ± 0.5) was used as standard to calculate FFU equivalents.

All analyses were performed on a 7500 Real-Time PCR System (ThermoFisher Scientific, Waltham, MA, USA).

### 2.5. TCID_50_ Determination

Infectious titers in form of a TCID_50_ were determined by limiting dilution as previously described [26,28]. Briefly, 2.5 × 10^4^ Vero E6 cells/well were seeded in 100 µL supplemented OptiPRO™ in a 96-well microtiter plate and incubated for 24 h before infection. Three days after infection, cells were washed once with PBS, followed by fixation with 4% paraformaldehyde in PBS for 20 min. After washing with PBS, cells were permeabilized with 0.5% TritonX100 in PBS for 15 min and blocked with 5% skimmed milk in PBS for 60 min. Subsequently, cells were stained with protein G purified sera from a convalescent patient, diluted 1:100 in PBS containing 2% skimmed milk for 60 min, followed by washing with PBS and incubation with goat anti-human IgG FITC (Jackson ImmunoResearch, West Grove, PA, USA #109-096-088) antibody. Cells were washed extensively and detection of positive cells was performed and the he signal was analyzed with the ImmunoSpot^®^ fluoro-X™ suite (Cellular Technology Limited, Cleveland, OH, USA) and TCID_50_ were calculated as described previously [26,29].

## 3. Results

### 3.1. Analysis of Vero E6 Closed Culture Conditions

In the CliniMACS Prodigy^®^, all required reagents and media are introduced into the single-use tubing set by sterile welding, allowing for surrounding-independent culture conditions (Figure A1). In the context of CAR-T cell therapy, this device offers a track record of high reproducibility and several cGMP-compliant protocols [30,31,32]. Non-GMP Research Cell Bank Vials of Serum-Free Vero-E6 Cells of documented WHO origin (ECACC no 88020401) were obtained from Nuvonis (Vienna, Austria) and cultured under serum-free conditions according to the provided instructions to generate a secondary research working cell bank. Starting with a freshly thawed aliquot of the secondary working cell bank, serum-free culture conditions in the CentriCult™ Unit (CCU) of the CliniMACS Prodigy^®^ were established. For this, the CCU was coated manually either with 30 mL of 5 µg/mL Biolaminin or with 30 mL PBS for 3 h at 37 °C, respectively. Subsequently, supernatant was discarded and 4.5 × 10^6^ freshly thawed Vero E6 cells were added to each CCU and incubated at 37 °C for up to 4 days with a daily media change and optical control of cell proliferation. As evident in Figure 1, the coating of the CCU greatly enhanced visible proliferation most likely due to better attachment of the cells to the CCU surface. Here, a confluent cell layer could be detected 48 h after inoculation, while confluency was not achieved in the uncoated CCU even after 96 h of cultivation. Accordingly, coating was performed prior to all subsequent experiments.

### 3.2. Establishing the Infection Protocol

In order to establish the programming of an infection workflow in a BSL-2 setting with the CliniMACS Prodigy^®^ (Figure A1), an initial infection experiment with the live-attenuated influenza vaccine Fluenz^®^ Tetra was performed. After installing the tubing set, the CCU was coated with Biolaminin and cells were added and cultured for 48 h at 39 °C (corresponding to 37 °C in the CCU) in the CliniMACS Prodigy^®^. After 48 h, infection was performed with 10^6^ FFU (per strain) in 30 mL OptiPRO™, corresponding approximately to a multiplicity of infection (MOI) of 0.1. After 4 h at 35 °C (33 °C in the chamber) media was changed to 30 mL fresh OptiPRO™ and a sample of the removed supernatant was taken. This step was repeated every 24 h for the following 5 days. Reverse transcriptase qPCR analysis of purified RNA from the respective samples revealed a minor drop followed by an increase in both IVA and IVB-specific RNA copies up to 3 days (IVA) or 4 days (IVB) post infection, indicating successful infection and virus propagation in the CCU (Figure 2).

### 3.3. Propagation of SARS-CoV-2

Having demonstrated the expansion of influenza viruses under closed cell culture conditions, the established workflow was adapted to the temperatures required for SARS-CoV-2 propagation and applied in subsequent infection experiments with a SARS-CoV-2 isolate in a BSL-3 laboratory. A virus isolate from the first passage of a COVID-19 patient was expanded by two additional passages under serum-free conditions in conventional cell culture flasks and aliquots of the filtered supernatants of the second passage were stored at −80 °C as virus working stocks.

Vero E6 cells were thawed, seeded and cultured for 48 h in a coated CCU in serum-free medium as described above. Cells were then infected with a MOI of 0.01 of the SARS-CoV-2 working stock for 4 h.

Subsequently, the media was automatically replaced with 30 mL fresh OptiPRO™ every 24 h and a sample of the removed media was stored for subsequent analyses. Automatic photographs taken by the integrated camera of the CliniMACS Prodigy system revealed the detachment of cells starting 2 days after infection (Figure 3). On day 3 after infection, cells were nearly entirely detached. In contrast, uninfected cells cultured as a control outside the CliniMACS Prodigy^®^ system mostly remained attached to the CCU for up to 4 days post infection.

Reverse transcriptase qPCR analysis of the sampled cell culture supernatants revealed peak viral RNA levels 3 days after infection (Figure 4A). Similarly, the infectious titer also increased more than 100-fold during this 3-day culture period (Figure 4B). To explore the consistency of the expansion of SARS-CoV-2 in the closed cell culture system, the infection experiment was repeated 4 and 6 weeks after the initial experiments. Viral RNA levels and infectious titers up to 3 days after infection differed by less than 14.6%/10.9%, respectively, demonstrating a striking reproducibility of this cell culture experiment up to 3 days after infection (Figure 4). Compared to the inoculation dose, an approximately 500-fold increase in viral RNA copies in the supernatants 3 days post infection was detected, indicating an efficient production of viral particles. As shown in Figure 4B, infectious titers peaked 3 days after infection with an approx. 300-fold increase in TCID_50_ values for all samples. These data highlight the possibility of efficiently and reproducibly propagating SARS-CoV-2 in a closed cell culture device.

## 4. Discussion

In this proof-of-concept study, we observed an approximately 300-fold amplification of SARS-CoV-2 in a closed cell culture system under serum-free conditions with high lot-to-lot consistency in the infectious titers obtained. Although the experiments were not performed under GMP conditions, we adhered to critical parameters such as the use of a well-defined producer cell, a serum-free virus working stock and serum-free culture conditions. All reagents, the cells, the medium and the virus inoculate were introduced by sterile tub welding into a single-use tubing set connected to the cell culture chamber. Similarly, the harvested supernatants were also removed by sterile tub welding, further ensuring that the viral preparations remain entirely separated from the environment of the biosafety level 3 laboratory. Of note, all steps in the closed cell culture Prodigy system were automated including the coating of the plates, the seeding of the cells, the centrifugation steps and the harvesting and filtration of the supernatants.

Based on the results of our proof-of-concept study, the following GMP-like production strategy for stocks of live-attenuated vaccines or challenge viruses under biosafety level 3 conditions seems feasible (Figure 5):A serum-free working cell bank of the producer cell and all reagents are prepared and stored in a conventional GMP laboratory.A clone of a virus isolate is passaged under serum-free conditions in the producer cell from the GMP laboratory using conventional cell culture flasks and the reagents from the GMP laboratory in the biosafety level 3 laboratory to generate a seed virus stock. The seed virus stock is aliquoted in the BSL-3 laboratory in a compounding aseptic containment isolator and stored under controlled freezing conditions.To generate a virus working stock, an aliquot of the seed virus stock is expanded using the automated closed cell culture system and the producer cell and reagents from the GMP laboratory. The virus working stocks are aliquoted and stored as described for the seed virus stocks.The virus working stocks are used in the closed cell culture system to generate different lots of the investigational live-attenuated vaccine or the challenge virus stock.The filling of the investigational products could be performed in a compounding aseptic containment isolator also located in the BSL-3 laboratory. Since live-attenuated vaccines or challenge viruses will most likely be administered by a mucosal rather than a parenteral route, purification by filtration seems sufficient, although additional purification steps could be implemented within the compounding aseptic containment isolator. The aliquoted samples are stored under controlled freezing conditions in the BSL-3 laboratory until use in the clinical study.

This production process for live-attenuated vaccines or challenge viruses for early clinical studies also seems feasible from a quantitative point of view if the administration of 10^5^ infectious doses of the live-attenuated SARS-CoV-2 vaccine candidate or the challenge virus is sufficient. Titers of up to 3 × 10^5^ infectious units/mL were obtained in the closed cell culture system (Figure 4B). As the cell culture unit included in the tubing set only offers 100 cm^2^ of culture surface, the total volume harvested was 30 mL. However, the possibility to attach external culture flasks by sterile welding seems an intriguing alternative. Connecting three 5-stack culture chambers, each with a 3180 cm^2^ cell growth area, could increase the usable cell growth area almost 100-fold. Thus, production volumes of 3 L per batch, corresponding to 3000 doses, seem possible. Such a number of doses should be sufficient for human challenge studies or for phase-I studies with live-attenuated vaccines. Once the tolerability and initial safety data are obtained for the live-attenuated vaccines, their biosafety level could be lowered, facilitating subsequent production of this live-attenuated vaccine under conventional GMP conditions.

In summary, this proof-of-concept study outlines a potential fast-track approach for the production of live-attenuated vaccine candidates based on highly pathogenic viruses under GMP-like conditions that may contribute to pandemic preparedness.

## Figures and Tables

**Figure 1 viruses-15-00397-f001:**
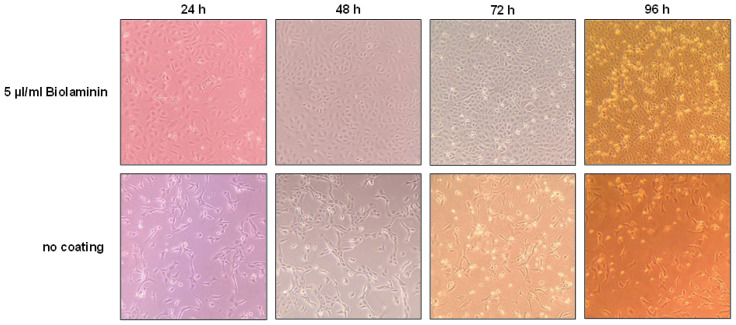
Establishing Vero-E6 cell culture conditions in the CliniMACS Prodigy CentriCult™ Units (CCU) were either coated with 30 mL 5 µg/mL Biolaminin in PBS or 30 mL PBS for 3 h at 37 °C. Subsequently, 4.5 × 10^6^ freshly thawed Vero-E6 cells were added to each CCU and incubated at 37 °C for 4 days. A picture was taken every 24 h, followed by a media change to 30 mL fresh glutamine-supplemented OptiPRO.

**Figure 2 viruses-15-00397-f002:**
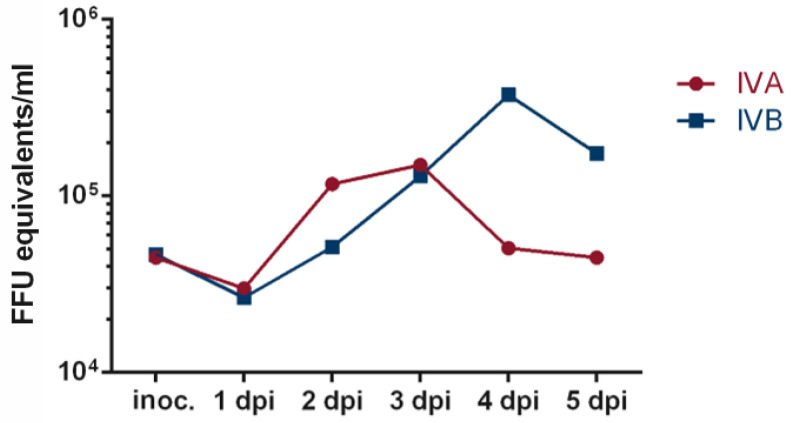
Fluenz^®^ Tetra infection of Vero-E6 cells. A CCU was coated with laminin in the CliniMACS Prodigy^®^ and subsequently seeded with 4.5 × 10^6^ freshly thawed Vero-E6 cells. After a 48-h incubation period at 39 °C, the culture was infected with Fluenz^®^ Tetra at a dose of 10^6^ FFU per strain. Media was changed and a sample taken after 4 h at 35 °C (corresponding to 33 °C in the CCU) followed by a daily media change and sample collection. Purified RNA from the samples was analyzed by Influenza A (IVA) and Influenza B (IVB)-specific reverse transcriptase qPCR. Purified RNA from Fluenz^®^ Tetra was assigned the FFUs contained in the volume of Fluenz^®^ Tetra the RNA was extracted from and used as a standard. Shown are FFU equivalents/mL for individual samples for IVA and IVB, respectively.

**Figure 3 viruses-15-00397-f003:**
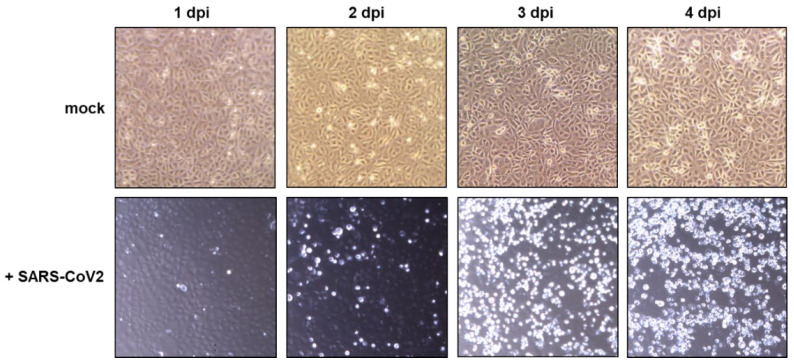
Optical comparison of mock and SARS-CoV-2 infected VeroE6 cells. An amount of 4.5 × 10^6^ freshly thawed Vero-E6 cells were seeded in a laminin-coated CCU and incubated at 37 °C for 48 h. For mock infection, media was changed every 24 h and a microscopic picture was taken. For SARS-CoV-2 infection, cells were infected with an MOI of 0.01 for 4 h, after which media was changed every 24 h and an automatic picture was taken every 12 h. Shown are representative pictures for matched time points as indicated. All steps of the SARS-CoV-2 infection were performed within the CliniMACS Prodigy, while the mock control cultures were maintained without CliniMACS Prodigy. Dpi: days post infection.

**Figure 4 viruses-15-00397-f004:**
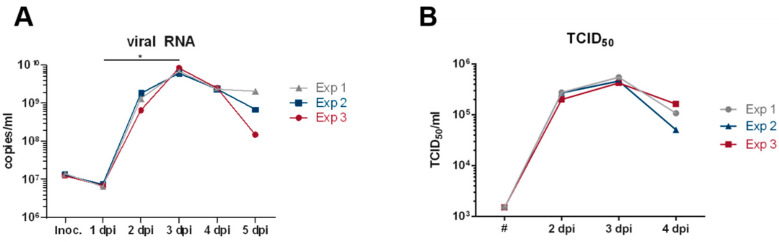
Analysis of SARS-CoV-2 propagation in the CliniMACS Prodigy^®^. An amount of 4.5 × 10^6^ freshly thawed Vero-E6 cells were seeded in a laminin-coated CCU. After 48 h of incubation at 39 °C (corresponding to 37 °C in the chamber) cells were infected with SARS-CoV-2 at a MOI of 0.01. After 4 h, media was changed and a sample taken (Inoc.) followed by cultivation for 5 days. During the culture period, media was exchanged and a sample taken every 24 h. (**A**) Purified RNA was analyzed by reverse transcriptase qPCR in duplicates and mean values for the respective samples are depicted from three independent experiments. For statistical analysis, Friedmans’ test with Dunn’s multiple comparison between all groups with log-transformed values was performed (* *p* < 0.05). (**B**) The infectious titers (TCID_50_) of samples of the three independent experiments on the indicated days post infection (dpi) are shown. # The TCID_50_ of the culture medium after inoculation was calculated from the TCID_50_ of the SARS-CoV-2 working stock and the applied dilution.

**Figure 5 viruses-15-00397-f005:**
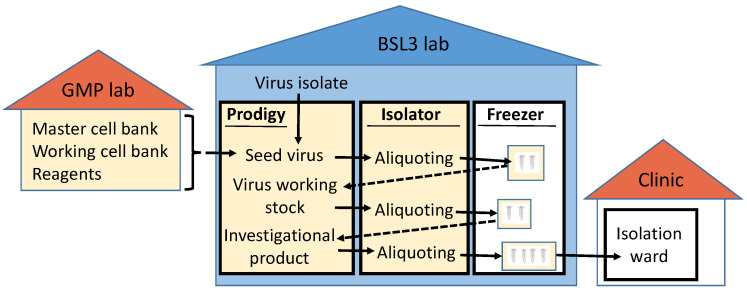
Concept for production of live-attenuated vaccines of SARS-CoV-2 under biosafety level 3 conditions. GMP-compatible working steps are shown in yellow boxes; BSL-3-containment is indicated by the blue background.

## Data Availability

Not applicable.

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
