# Peer review of "Propagation of SARS-CoV-2 in a Closed Cell Culture Device: Potential GMP Compatible Production Platform for Live-Attenuated Vaccine Candidates under BSL-3 Conditions?"

_viruses, 2023, doi:10.3390/v15020397_

Round 1
Reviewer 1 Report
I do not think there is not enough innovation in this study. There are much improvement need to qualify the production of virus.
Author Response
Response:
To our knowledge, this is the first time a closed cell culture system is used to propagate viruses. We therefore feel, that the manuscript is very focused, but also innovative. The comment of the other reviewer confirms our impression, that virologist are rarely aware of the possibility to propagate viruses in closed cell culture systems.

Reviewer 2 Report
Major Concerns
1. The eFFU/ml numbers at the inoculum timepoint in Figure 2 don’t seem to correspond to the methods. You seem to be reporting approximately 5x105 FFU equivalents/ml for IVA and IVB at this timepoint. However, you state that you infected the flask with a total of 106 FFU of each strain in a volume of 30ml (Lines 192-193), and that this inoculum was removed after 4 hours and replaced with 30ml of fresh media and a sample of the supernatant was taken (Lines 194-195). Taken alone, I would interpret this as meaning that your sample was collected from that 30ml of fresh media that was added after the 4 hour infection, in which case the titer should be extremely low. However, based on your description of the SARS-CoV-2 infections I think you’re saying that this titer is reflective of the removed inoculum, not the replaced media (in which case make sure that this is more clearly written). In this case, your titer shouldn’t be any higher than 106 FFU/30ml=3x104 FFU/ml, and that’s not accounting for any virus that either entered the cells or became non-infectious over the course of the 4 hour infection. Is it possible that there was an error in converting ct values to FFU/ml equivalents? Double check your raw and calculated data and/or update the description of your method to ensure accurate reporting.
2. The inoculum titers reported in 4A and 4B appear to be absolutely identical between the three independent experiments. Given that the 14.6% variability in viral RNA and 10.9% variability in infectious virus at day 3 was visible on the scale of your graphs, and I can even see hints of some quite small variability in RNA at 1 dpi, please confirm that the RNA and infectious titer reported in the inoculum reflects experimental results and not just the expected input based on your stock titers.
Minor Concerns
1. As this manuscript focuses on the utility of the CliniMACS Prodigy system, more detailed information on its construction and capabilities would be helpful. I was able to find what I needed online, but including an image or diagram of the CliniMACS unit in Results Section 3.1 where you describe the system (Lines 161-165) would be helpful.
2. Regarding the SARS-CoV-2 strain used in these experiments, you state that it is a Vero E6 p1 isolate from a patient in Erlangen and cite reference 26 (Peter et al) for further details. However, the cited paper does not provide the pertinent details of virus isolation or strain information, at least not in an easily accessible way. It would be preferable to provide any available information regarding the strain name, sequence accession number, or lineage/variant classification directly in your methods section.
3. You do not mention influenza in the methods section other than to describe your qPCR assay. Include the source information and propagation (if performed) for the Fluenz Tetra. As the infection is reported in FFU and the qPCR results are reported as FFU equivalents, I assume that either (a) the vaccine came to you with an FFU titer listed and this should be specified when describing the virus in the methods, or (b) you performed focus forming assays and should include this protocol in the methods.
4. Include appropriate statistical analysis for the growth curves in Figure 4 comparing the inoculum titers vs. the titers at later timepoints to state which timepoints represent a significant amount of viral replication compared to the inoculum. I would suggest that something akin to a either a repeated measures ANOVA or Friedmans’ test performed on log10-tranformed titers is likely appropriate.
5. The authors indicate in Lines 311-313 that the biosafety level for a live-attenuated vaccinate candidate can be lowered once initial human safety data is obtained. However, it is possible to lower the biosafety level of an attenuated virus such as a vaccine candidate based on in vitro and animal safety data, depending on the regulatory framework of the institution and country. It is therefore not necessarily true that a live attenuated vaccine candidate for a BSL-3 virus would have to be manufactured in a BSL-3 laboratory if sufficient preliminary data exists (as would presumably be the case for a vaccine candidate about to enter human trials). The tone of the introduction and discussion should thus be softened somewhat to acknowledge that this platform has clear utility in the production of challenge stocks and in some, but not in all, initial production runs for live attenuated vaccine candidates.
Typographical Edits
1. Add a hyphen between SARS and CoV-2 in the title (Line 2)
2. Change ‘themselves as promising’ to ‘themselves as a promising’ (Line 14)
3. Change ‘induction of a broad mucosal’ to ‘induction of broad mucosal’ (Line 15)
4. Change 40.000 to scientific notation (Line 79)
5. Change CO2 to CO2 (Line 114)
6. Remove the first sentence in section 3.1 “As automate…Prodigy was used.” (Line 161)
7. Replace the word nonhazardous with a more precise description such as BSL-2 (Line 188)

Author Response
Major Concerns
- The eFFU/ml numbers at the inoculum timepoint in Figure 2 don’t seem to correspond to the methods. You seem to be reporting approximately 5x105 FFU equivalents/ml for IVA and IVB at this timepoint. However, you state that you infected the flask with a total of 106 FFU of each strain in a volume of 30ml (Lines 192-193), and that this inoculum was removed after 4 hours and replaced with 30ml of fresh media and a sample of the supernatant was taken (Lines 194-195). Taken alone, I would interpret this as meaning that your sample was collected from that 30ml of fresh media that was added after the 4 hour infection, in which case the titer should be extremely low. However, based on your description of the SARS-CoV-2 infections I think you’re saying that this titer is reflective of the removed inoculum, not the replaced media (in which case make sure that this is more clearly written). In this case, your titer shouldn’t be any higher than 106 FFU/30ml=3x104 FFU/ml, and that’s not accounting for any virus that either entered the cells or became non-infectious over the course of the 4 hour infection. Is it possible that there was an error in converting ct values to FFU/ml equivalents? Double check your raw and calculated data and/or update the description of your method to ensure accurate reporting.
Response:
As noted by the reviewer the inoculation sample was taken from the supernatant that was removed 4 hours after infection. To make this clearer we added “removed” in line 207. Also, after thoroughly checking the raw data we detected a copy and paste error of the standard between different softwares used, leading to a 10-fold inflation of all values. The corrected values are now displayed in Fig. 2. (corresponding to ~4.5*104 FFU equivalents, therefore within the expected value when calculating from stock concentrations of 5*107±0.5 FFU/ml (per strain) as given by the manufacturer)
Additionally, (in accordance with minor concern 3) we added a description for the propagation of Influenza Virus under Section 2.3 in Materials and Methods to more thoroughly describe the Influenza Propagation in a closed cell culture device.
- The inoculum titers reported in 4A and 4B appear to be absolutely identical between the three independent experiments. Given that the 14.6% variability in viral RNA and 10.9% variability in infectious virus at day 3 was visible on the scale of your graphs, and I can even see hints of some quite small variability in RNA at 1 dpi, please confirm that the RNA and infectious titer reported in the inoculum reflects experimental results and not just the expected input based on your stock titers.
Response:
The qPCR results shown in Fig 4A for the time point “inoc.” were experimentally determined and showed very low variability. For Fig 4B the working stock titer was experimentally determined and the TCID50 of the culture medium after inoculation was calculated from the TCID50 of the working stock and the dilution applied. We now adjusted the wording in the figure and the corresponding figure legend to more accurately describe the respective data.
Minor Concerns
- As this manuscript focuses on the utility of the CliniMACS Prodigy system, more detailed information on its construction and capabilities would be helpful. I was able to find what I needed online, but including an image or diagram of the CliniMACS unit in Results Section 3.1 where you describe the system (Lines 161-165) would be helpful.
Response:
We added a schematic overview of the CliniMACS Prodigy as Figure A1 in the appendix and referenced it in the appropriate sections throughout the manuscript.
- Regarding the SARS-CoV-2 strain used in these experiments, you state that it is a Vero E6 p1 isolate from a patient in Erlangen and cite reference 26 (Peter et al) for further details. However, the cited paper does not provide the pertinent details of virus isolation or strain information, at least not in an easily accessible way. It would be preferable to provide any available information regarding the strain name, sequence accession number, or lineage/variant classification directly in your methods section.
Response:
We added strain name, and sequence accession number in the methods section (2.2.) , as well as a remark to the supplementary information of reference 26.
- You do not mention influenza in the methods section other than to describe your qPCR assay. Include the source information and propagation (if performed) for the Fluenz Tetra. As the infection is reported in FFU and the qPCR results are reported as FFU equivalents, I assume that either (a) the vaccine came to you with an FFU titer listed and this should be specified when describing the virus in the methods, or (b) you performed focus forming assays and should include this protocol in the methods.
Response
Additionally to the in Major concern 1 mentioned addition of Fluenz Tetra propagation, a description of FFU equivalents calculation was added to the Methods section “2.4 qPCR analysis”.
- Include appropriate statistical analysis for the growth curves in Figure 4 comparing the inoculum titers vs. the titers at later timepoints to state which timepoints represent a significant amount of viral replication compared to the inoculum. I would suggest that something akin to a either a repeated measures ANOVA or Friedmans’ test performed on log10-tranformed titers is likely appropriate.
Response
Friedmanns Test was performed for qPCR results as suggested by the reviewer.
For TCID50 values, no statistical analysis could be performed, as no experimental data for the individual inoculation titers were determined (see major concern 2)
- The authors indicate in Lines 311-313 that the biosafety level for a live-attenuated vaccinate candidate can be lowered once initial human safety data is obtained. However, it is possible to lower the biosafety level of an attenuated virus such as a vaccine candidate based on in vitro and animal safety data, depending on the regulatory framework of the institution and country. It is therefore not necessarily true that a live attenuated vaccine candidate for a BSL-3 virus would have to be manufactured in a BSL-3 laboratory if sufficient preliminary data exists (as would presumably be the case for a vaccine candidate about to enter human trials). The tone of the introduction and discussion should thus be softened somewhat to acknowledge that this platform has clear utility in the production of challenge stocks and in some, but not in all, initial production runs for live attenuated vaccine candidates.
Response
The introduction was rephrased to reflect the possibility of downgrading of the biosafety level of live attenuated vaccine candidates as suggested by the reviewer.
Typographical Edits
- Add a hyphen between SARS and CoV-2 in the title (Line 2)
- Change ‘themselves as promising’ to ‘themselves as a promising’ (Line 14)
- Change ‘induction of a broad mucosal’ to ‘induction of broad mucosal’ (Line 15)
- Change 40.000 to scientific notation (Line 79)
- Change CO2 to CO2 (Line 114)
- Remove the first sentence in section 3.1 “As automate…Prodigy was used.” (Line 161)
- Replace the word nonhazardous with a more precise description such as BSL-2 (Line 188)
Response
All typographical edits were implemented in the manuscript
